Galaxy tools and workflows for sequence analysis with applications in molecular plant pathology

Cock Peter J.A. 1 peter.cock@hutton.ac.uk
Grüning Björn A. 2
Paszkiewicz Konrad 3
Pritchard Leighton 1
1 Information and Computational Sciences, James Hutton Institute , UK
2 Pharmaceutical Bioinformatics, Institute of Pharmaceutical Sciences, Albert-Ludwigs-University , Freiburg , Germany
3 Wellcome Trust Biomedical Informatics Hub and Exeter Sequencing Service, University of Exeter , UK
Lazo Gerard
Electronic publication date: 2013 Sep 17
Publication date: 2013
Volume: 1
Electronic Location ID: e167
Received 2013 Jul 17; Accepted 2013 Aug 30
Copyright: © 2013 Cock et al.
Copyright year: 2013
Copyright holder: Cock et al.
License: This is an open access article distributed under the terms of the Creative Commons Attribution License, which permits unrestricted use, distribution, and reproduction in any medium, provided the original author and source are credited.
License URL: https://creativecommons.org/licenses/by/3.0/

Keywords: Galaxy, Pipeline, Accessibility, Effector proteins, Workflow, Reproducibility, Annotation, Sequence analysis, Genomics

Funding: Scottish Government Rural and Environmental Research and Analysis Directorate Excellence Initiative of the German Federal and State Governments through the Junior Research Group Program (ZUK 43) PJC and LP are supported by the Scottish Government Rural and Environmental Research and Analysis Directorate. The work from BAG has been funded by the Excellence Initiative of the German Federal and State Governments through the Junior Research Group Program (ZUK 43). The funders had no role in study design, data collection and analysis, decision to publish, or preparation of the manuscript.

==============================
The Galaxy Project offers the popular web browser-based platform Galaxy for running bioinformatics tools and constructing simple workflows. Here, we present a broad collection of additional Galaxy tools for large scale analysis of gene and protein sequences. The motivating research theme is the identification of specific genes of interest in a range of non-model organisms, and our central example is the identification and prediction of “effector” proteins produced by plant pathogens in order to manipulate their host plant. This functional annotation of a pathogen’s predicted capacity for virulence is a key step in translating sequence data into potential applications in plant pathology.

This collection includes novel tools, and widely-used third-party tools such as NCBI BLAST+ wrapped for use within Galaxy. Individual bioinformatics software tools are typically available separately as standalone packages, or in online browser-based form. The Galaxy framework enables the user to combine these and other tools to automate organism scale analyses as workflows, without demanding familiarity with command line tools and scripting. Workflows created using Galaxy can be saved and are reusable, so may be distributed within and between research groups, facilitating the construction of a set of standardised, reusable bioinformatic protocols.

The Galaxy tools and workflows described in this manuscript are open source and freely available from the Galaxy Tool Shed (http://usegalaxy.org/toolshed or http://toolshed.g2.bx.psu.edu).

Introduction

Biological motivation

Crop plants are constantly exposed to environmental challenges, including attack by microbial pathogens, pests and parasites such as bacteria, fungi, oomycetes, nematodes and insects. These interactions affect product quality and yield and result in billions of dollars’ worth of crop losses worldwide each year (Newton et al., 2010). Understanding the biochemical actions of these pathogens, and the responses of their hosts, is a key goal of plant pathology that is expected to enable improved disease control strategies through plant breeding, engineering, and chemical methods. Plants have evolved an array of defence mechanisms including passive physical barriers and active defence responses such as the production of reactive oxygen species, and programmed cell death. Successful microbial pathogens must evade or interfere with these defences and, in order to do so, many are able to introduce molecules called effectors into host plant cells or their immediate environment (Fig. 1) (Hématy, Cherk & Somerville, 2009; Stergiopoulos & de Wit, 2009; Dodds & Rathjen, 2010; Hann, Gimenez-Ibanez & Rathjen, 2010; Rico, Mccraw & Preston, 2011). Effectors interact with the plant’s biochemistry and may subvert its defences or otherwise modify host behaviour to produce a more favourable environment for the pathogen. The identification of a pathogen’s effector complement and prediction of their biochemical function is key to delineating an individual pathogen’s capacity for virulence, and motivates this work. The wider study of effectors and their targets in the host is likely also to be instructive for understanding general principles of pathogenicity (Schneider & Collmer, 2010; Pritchard & Birch, 2011).

Figure 1 Schematic of effector infiltration into a host plant cell.

Plant pests and pathogens introduce effector proteins into the host plant cell (green), where they can target and manipulate plant biochemistry to the benefit of the pathogen (Dodds & Rathjen, 2010). Effectors may be delivered by haustorial ingression from a fungus or oomycete such as P. infestans (orange), via the bacterial Type III secretion system mechanism (blue), by nematodes via injection into the plant cell though a needle-like stylet (red), or many other processes (not illustrated). Where effectors may be identified by sequence properties, candidate effector proteins can be computationally predicted using Galaxy.

Many known effector proteins can be considered to have a modular structure comprising an ‘address’ domain responsible for directing the effector to a particular location, and a ‘message’ domain containing a biochemically functional ‘payload’. Where these domains can be reliably identified on the basis of their sequence or genomic context, effector prediction on a genome-wide scale is possible by computational methods. For example, cytoplasmic effectors that act within host cells must be transferred from the pathogen into the plant (Fig. 1). The translocation mechanism may involve specific sequence properties, or a sequence motif, as in the bacterial Type III secretion system (Cunnac, Lindeberg & Collmer, 2009; Arnold et al., 2009; Jehl, Arnold & Rattei, 2011), and for oomycete RXLR effectors (Whisson et al., 2007; Block et al., 2008; Birch et al., 2009). A choice of bioinformatic methods, with differing specificity and sensitivity, exists for the identification of these motifs (Haas et al., 2009; Wang et al., 2011; Jehl, Arnold & Rattei, 2011). The host targets of most effector proteins are not yet known, which is problematic for computational prediction or identification of effector functional domains. A notable exception is the Xanthomonas AvrBs3 (also known as TAL) family of effectors that act by binding DNA and regulating host gene expression (Moscou & Bogdanove, 2009; Boch et al., 2009; Bogdanove, Schornack & Lahaye, 2010). Other effectors may be identifiable by generic characteristics such as specific protein domains, or sequence similarity to known effectors.

The ability to generate rapidly large amounts of genomic and transcriptomic sequence data for non-model organisms has enabled comparative genomic studies of a large number of plant pathogen effectors (Haas et al., 2009; Baltrus et al., 2011). Prior to the commonplace production of genome scale data, manual workflows for functional characterisation of effector sequences, often involving copying and pasting sequences into online tools like NCBI BLAST (Altschul et al., 1990; Camacho et al., 2009) were tractable. These labour-intensive approaches are impractical with large datasets, for which automated large-scale analyses become necessary. Adopting an automated workflow also brings benefits as, even when the level of data would be manageable, manual analyses can be difficult to reproduce without meticulous record keeping. This can affect the consistency of work within a research group, and also the utility of published literature, where the level of detail in the computational methods section can be inadequate for replication. Automated analyses are usually repeatable, and both the analytical processes and results can be logged in great detail, in a searchable framework.

This manuscript focuses on the analysis of whole organism gene sets containing thousands of genes, and the assembly and automated annotation steps that produce them. Dealing with high throughput sequencing data was also one of the motivations for the original development of the Galaxy Project (Goecks et al., 2010; Blankenberg et al., 2010).

Technological motivation

There are dozens, if not hundreds, of commonly-used tools for sequence analysis, such as NCBI BLAST and the EMBOSS suite (Rice, Longden & Bleasby, 2000), and they are often applied in concert as a workflow. Traditionally, this was achieved by chaining together command line tools and online resources with short scripts. Typically, the bash (or other) shell, or languages such as Perl or Python are used, often with libraries like BioPerl (Stajich et al., 2002) or Biopython (Cock et al., 2009) which include parsers and wrappers for many command line tools. However, acquiring the skill-set for such practical bioinformatics requires a time investment which may be a burden to the active wet-lab researcher. Additional to the obvious requirement of learning to program, the user is typically required to work at the command line prompt, and on Linux or Apple’s OS X operating system rather than Microsoft Windows (which may be more familiar). It’s also likely that basic systems administration skills will be needed to install and update the underlying tools and databases. Ideally, the user would also possess software engineering skills to identify and resolve bugs, and be able to apply professional concepts such as unit tests and version control. While some biological scientists from a laboratory background master these skills, they are, in our experience, the exception rather than the rule. Creation of automated analyses has therefore so far remained largely a specialised niche, limiting their wider uptake and application.

One reason for NCBI BLAST’s wide uptake is that it is easy to use as it can be accessed over the internet via any recent web-browser. Many groups that develop new analysis tools offer them via a web-browser interface, which incurs additional effort for the tool developer, and requires maintenance of computer resources. The instant availability of the web tools to potential users can encourage uptake (and thus, indirectly, citations), but in the absence of a web application programming interface (API) or web service, online tools may not be readily integrated with each other, or with other software. Even in the case where an API exists, and the output of Tool A is in a format understood by Tool B, for a multi-step analysis the user would still have to move the data between the two services for each analysis.

The solution that Galaxy offers is multiple tools or data resources that can be linked together using a common, user-friendly interface. Under this model, the end user interacts with a single web server, where they can run the individual tools they need via a single common interface, but where they are also able to couple tools together, making the output of one tool the input of another. These multi-step analyses may be saved as workflows which can then be repeated on other datasets, and optionally shared with colleagues to use on their own datasets. Building a biologically relevant and useful workflow using Galaxy still requires insight into the problem at hand, the nature of the data and the available tools, but it no longer requires a high level of computing skill. Also, sharing and reuse of existing workflows becomes very straightforward, improving reproducibility and reusability of analyses within and between research groups.

Galaxy

The Galaxy software runs on Linux/Unix based servers, and provides a browser-based user interface (see for example Fig. 2). The end user can access Galaxy from any computer operating system with a recent web browser (including Mozilla Firefox, Apple Safari, Google Chrome and Microsoft Internet Explorer). Individual tools offered via Galaxy are installed and run on the Galaxy server or associated computing cluster, meaning the end user does not have to download or install the tools themselves. This avoids numerous practical problems with deployment and update of tools to individual desktops, many of which are complicated to install, may differ between versions, rely on unstated dependencies, or may not even run on the desktop operating system. There is an additional usability benefit to this presentation, as all tools provided via Galaxy are given a consistent and familiar graphical user interface. This contrasts with the alternative, diverse interfaces encountered when working with command line tools via the keyboard, or the many different tool-specific web-services, each with different interfaces and account settings.

Figure 2 Screenshot of the Galaxy PSORTb v3.0 wrapper.

The left hand pane (A) holds a menu of tools which is configurable by the Galaxy administrator. The central pane (B) shows the currently selected tool or dataset, here PSORTb. The right hand pane (C) holds the current datasets, and is empty in this example. The tool interface presents the user with familiar drop down list controls (in this example a file selector and other parameters), option radio-selectors, check boxes, or text boxes as defined in the tool configuration file. For PSORTb, text input boxes are restricted to only accept numeric values. Tool input parameters are followed by a blue “Execute” button that runs the tool when clicked. Below this, user documentation and citation information are provided.

The Galaxy developers host a public Galaxy server http://usegalaxy.org at Pennsylvania State University (PSU), which offers a broad range of installed tools. Galaxy can also be downloaded and run on a local server, which is a route followed by many research institutes, departments and even individual research groups. This is useful for researchers working on ethically or commercially sensitive data, for whom uploading to the PSU (or any other public) server may not be possible. The upload of large datasets to a public server may also be limited by available bandwidth and user quotas, and a local installation may also be useful in this case. Local Galaxy instances can be configured to run jobs on a local computing cluster, which may also return results more rapidly than the increasingly popular public servers. Access to local Galaxy servers can be restricted to the local intranet, or the server can be made available over the internet (e.g., if necessary for collaboration, or to make a tool available to the scientific community). Finally, Galaxy “CloudMan” can also be run on a “Cloud Computing” platform using machines rented from a provider such as Amazon Inc., which offers an easy way to enlarge the rented computer cluster based on usage demands (Afgan et al., 2010).

From our perspective, the key benefit of a local or cloud Galaxy server over using a public instance is the increased level of control of the tool set offered. We are able to add tools of local interest, including software whose licence allows for use within the group, but not provision of a public service. All of the tool wrappers we describe in this manuscript (Table 1) are available on the “Galaxy Tool Shed” (http://usegalaxy.org/toolshed or http://toolshed.g2.bx.psu.edu), a public repository of tools and workflows developed and shared by the wider Galaxy community.

Table 1 Summary of Galaxy tools, wrappers for existing tools, and sample workflows discussed in this manuscript.

Some Galaxy tools are new pieces of software written specifically for use within Galaxy, others are wrappers allowing an existing tool to be used within Galaxy. Most of the tools wrapped are freely available under an open source license, however those marked ⋆ are proprietary but free for academic use only, while † indicates free to download but with unspecified terms. Galaxy workflows are saved recipes or pipelines which automate running one or more Galaxy tools. See Materials and Methods for more details.

Galaxy Tool Shed URL, description	Type	References	
http://toolshed.g2.bx.psu.edu/view/devteam/ncbi_blast_plus			
Standalone NCBI BLAST+ tools	Wrappers	Altschul et al. (1990);	
		Camacho et al. (2009)	
http://toolshed.g2.bx.psu.edu/view/devteam/blast_datatypes			
BLAST datatype definitions (BLAST XML, databases)	Datatypes	Altschul et al. (1990);	
		Camacho et al. (2009)	
http://toolshed.g2.bx.psu.edu/view/peterjc/blastxml_to_top_descr			
BLAST top hit descriptions	New tool		
http://toolshed.g2.bx.psu.edu/view/peterjc/blast2go			
Blast2GO for pipelines (b2g4pipe)	Wrapper†	Conesa et al. (2005);	
		Götz et al. (2008)	
http://toolshed.g2.bx.psu.edu/view/peterjc/mira_assembler			
MIRA assembler	Wrapper	Chevreux, Wetter & Suhai (1999)	
http://toolshed.g2.bx.psu.edu/view/bgruening/augustus			
Augustus, for eukaryotic gene finding	Wrapper	Keller et al. (2011)	
http://toolshed.g2.bx.psu.edu/view/bgruening/glimmer3			
Glimmer3, for prokaryotic gene finding	Wrappers	Delcher et al. (2007)	
http://toolshed.g2.bx.psu.edu/view/bgruening/repeat_masker			
RepeatMasker, for screening DNA sequences	Wrapper	Smit, Hubley & Green (1996–2010)	
http://toolshed.g2.bx.psu.edu/view/bgruening/interproscan			
InterProScan	Wrapper	Zdobnov & Apweiler (2001);	
		Quevillon et al. (2005)	
http://toolshed.g2.bx.psu.edu/view/peterjc/tmhmm_and_signalp			
SignalP, for signal peptide prediction	Wrapper⋆	Bendtsen et al. (2004)	
TMHMM, for trans-membrane domain prediction	Wrapper⋆	Krogh et al. (2001)	
PSORTb, for bacterial/archaeal proteins	Wrapper	Yu et al. (2010)	
WoLF PSORT, for fungi/animal/plant proteins	Wrapper⋆	Horton et al. (2007)	
Promoter, for eukaryotic PolII promoters	Wrapper⋆	Knudsen (1999)	
Oomycete RXLR motifs	New Tool	Bhattacharjee et al. (2006);	
		Win et al. (2007);	
		Whisson et al. (2007)	
http://toolshed.g2.bx.psu.edu/view/peterjc/predictnls			
PredictNLS, predict nuclear localization sequence	Rewrite	Cokol, Nair & Rost (2000)	
http://toolshed.g2.bx.psu.edu/view/peterjc/nlstradamus			
NLStradamus, a nuclear localization sequence predictor	Wrapper	Nguyen Ba et al. (2009)	
http://toolshed.g2.bx.psu.edu/view/peterjc/clinod			
NoD, nucleolar localization sequence detector	Wrapper†	Scott et al. (2010);
Scott, Troshin & Barton (2011)	
http://toolshed.g2.bx.psu.edu/view/peterjc/effectivet3			
EffectiveT3, predicts bacterial type III secretion signals	Wrapper	Arnold et al. (2009);
Jehl, Arnold & Rattei (2011)	
http://toolshed.g2.bx.psu.edu/view/peterjc/venn_list			
Venn Diagrams from (gene) identifier lists	New Tool		
http://toolshed.g2.bx.psu.edu/view/peterjc/seq_filter_by_id			
Filter sequences by (gene) identifier	New Tool		
http://toolshed.g2.bx.psu.edu/view/peterjc/seq_select_by_id			
Select sequences by (gene) identifier	New Tool		
http://toolshed.g2.bx.psu.edu/view/peterjc/seq_rename			
Rename identifiers in sequence files	New Tool		
http://toolshed.g2.bx.psu.edu/view/peterjc/fastq_paired_unpaired			
FASTQ deinterlacer for paired reads	New Tool		
http://toolshed.g2.bx.psu.edu/view/peterjc/get_orfs_or_cdss			
Open reading frame (ORF) and crude coding sequence (CDS) prediction	New Tool		
http://toolshed.g2.bx.psu.edu/view/bgruening/glimmer_gene_calling_workflow			
Glimmer gene calling with training-set	Workflow		
http://toolshed.g2.bx.psu.edu/view/peterjc/secreted_protein_workflow			
Secreted proteins using SignalP and THMHMM	Workflow	Cock & Pritchard (in press)	
http://toolshed.g2.bx.psu.edu/view/peterjc/rxlr_venn_workflow			
Venn Diagram comparison of oomycete RXLR predictions	Workflow	Cock & Pritchard (in press)	

Materials and Methods

Table 1 summarises a number of tools we have wrapped, or written, for use in Galaxy. Links are provided to their location in the Galaxy Tool Shed. Example usage is discussed in the Results and Discussion below.

The original Galaxy release did not include wrappers for the standalone NCBI BLAST or BLAST+ command line tools (Altschul et al., 1990; Camacho et al., 2009). The use of BLAST was a priority for our own work, so we developed wrappers for the core BLAST+ tools. These were initially included in the main Galaxy repository before being migrated to the Galaxy Tool Shed. The BLAST+ tools have not been made available at the http://usegalaxy.org public server due to concerns over the resulting computational load (J Taylor, pers. comm., 2013), but are pre-installed on Galaxy CloudMan images, and can easily be added to a local Galaxy installation.

The majority of the Galaxy tools we describe in this manuscript are wrappers around existing third-party tools. In many cases the relevant input and output file formats of those tools are standard bioinformatics data-exchange formats, such as FASTA for sequence data and plain text tabular output for numeric data, which were already defined within Galaxy. However, the outputs of many of the tools wrapped required reformatting into a plain text tab-separated table, which is the basic representation of data within Galaxy. The output from some tools, for example, the type III effector prediction package EffectiveT3, which produces semicolon-separated output (Arnold et al., 2009; Jehl, Arnold & Rattei, 2011), and SignalP v3.0, which produces space separated output (Bendtsen et al., 2004), required minor conversion to tab-separated output. Other tools, such as the nuclear localisation signal prediction tools NLStradamus (Nguyen Ba et al., 2009) and NoD (Scott et al., 2010; Scott, Troshin & Barton, 2011), produced no suitable output. In these cases, a tabular output option was added to each tool, following discussion with the authors. Some tools, such as NCBI BLAST+, can produce data in their own specific file formats. In these cases, new Galaxy datatypes can be defined, and this was done to enable wrappers for NCBI BLAST+. To support BLAST XML in particular, additional code was required to support job-splitting, automatic recognition of the filetype during import, and the option to convert BLAST XML into different tabular formats. Additional datatypes were also required to support protein and nucleotide BLAST databases.

Many of the underlying tools wrapped for Galaxy run in a single thread, using only one CPU at any one time. Some of the wrappers described in this manuscript attain a significant speedup relative to the standalone tool, by dividing the input data into batches and running a separate instance of the underlying tool, in parallel, on each batch of data. This process is completely transparent to the user, and allows the BLAST+ wrappers, for example, to specify that input FASTA query files should be broken up into batches of 1000 sequences, and the resulting BLAST output files merged afterwards. Distributing the input data in this way also provides opportunity for data sanitisation, such as the removal of extremely long FASTA description lines (which can cause some of the wrapped tools to fail), and avoids any hard coded limits on the number of sequences supported by some tools (e.g., SignalP v3.0 has a built in default limit of 4000 input sequences).

Wrapping of the nuclear localisation prediction tool PredictNLS (Cokol, Nair & Rost, 2000) was particularly problematic, as it lacked a batch mode allowing it to be used on FASTA files containing multiple query sequences. In order to implement an efficient solution, we reimplemented the core algorithm in Python (under the same open source licence), and have been in dialogue with the Rost Lab to incorporate these changes into the official PredictNLS package.

Some tools we describe were written from scratch as new Python scripts, rather than as a wrapper for an existing tool. One example in this manuscript reproduces three published methods for oomycete effector RXLR motif prediction (Bhattacharjee et al., 2006; Win et al., 2007; Whisson et al., 2007).

We found it useful to add further generic sequence manipulation tools to the standard Galaxy instance. These are written as Python scripts using the Galaxy and/or Biopython (Cock et al., 2009) modules. Our experience is that the sequence filtering tool has been of the greatest general utility. For example, the operation of dividing a multiple sequence file into those sequences which either do or do not have some property, such as a BLAST match, is a common task in many analyses.

In order to avoid “silent failures” where a tool might appear, within Galaxy, to have run but to have produced no output, special care was taken with error handling. Not all of the wrapped third-party tools follow Unix norms, and some fail to set an appropriate exit status code. In these cases, it could be impossible to determine whether the tool had failed, or had simply produced no output. This sometimes required wrapper code to handle special cases, such as when a run was aborted with no output produced. Where possible, unit tests have been included using the Galaxy framework. These tests are automatically run on the Galaxy Tool Shed for quality control, and should also be run whenever installing a Galaxy tool locally.

Installation instructions for each Galaxy tool are included in their documentation, which accompanies the tool itself on the Galaxy Tool Shed. The Galaxy Tool Shed framework offers the administrators of a local Galaxy installation point-and-click installation of new tools. In many cases, the underlying dependencies are handled automatically, greatly easing the process of tool installation. Automated installation is possible for most of the open source tools we describe in this manuscript, but where proprietary licenses apply, some dependencies must be installed manually. In particular, the third-party tools Promoter 2.0 (Knudsen, 1999), TMHMM 2.0 (Krogh et al., 2001), and SignalP 3.0 (Bendtsen et al., 2004) from the Center for Biological Sequence Analysis at the Technical University of Denmark, and WoLF PSORT (Horton et al., 2007) from the National Institute of Advanced Science and Technology (AIST), Japan, have licences that do not permit redistribution in this manner. Furthermore, running these tools on a public Galaxy server appears to be prevented under their limited free for academic use licenses.

Complex tools with a large number of options present a particular challenge to wrapper design, when attempting to balance the desire to offer full control and flexibility against usability. In the case of the NCBI BLAST+ tools, we chose initially to omit some of the less commonly-used options, and to provide others in an ‘advanced options’ section. Similarly, the wrapper for the MIRA assembler (Chevreux, Wetter & Suhai, 1999) currently exposes only the most common arguments as user-configurable parameters.

We also aimed to provide thorough end user documentation in the interface, at point of use. This is shown below the tool controls and provides guidance on typical usage, input and output file formats, and the relevant reference or citation information (e.g., Fig. 2B).

Results and Discussion

We now briefly outline several example workflows made possible by the tools and wrappers we describe in this manuscript. General tools for “Next Generation Sequencing” (NGS) are especially well served in Galaxy. However, the more specialised the tools become (typically, the further downstream your analysis), the less likely it is that a specific desired tool has already been wrapped for use in Galaxy. Although we have also implemented wrappers to facilitate basic genome assembly and gene calling, here we focus on what happens after assembly and gene calling has been performed, to answer the question “What can be learned from the predicted gene complement of a newly sequenced organism?”.

The ability to answer well this motivating question regarding protein function rests on the quality of basic annotation, for which tools such as BLAST, Blast2GO and InterProScan may be used. When a microbial plant pathogen is sequenced and a gene complement predicted, a particular aim of functional annotation is to identify candidate effector proteins. Typical methods employed include determination of sequence similarity to known effectors from related organisms (e.g., using BLAST), enhanced transcription in specific pathogen life- or infection stages (using microarrays, RNASeq, or quantitative PCR), and exploitation of the modular structure typically expected of effectors to identify novel effector candidates. Many of the tools described in Table 1 are used to identify these ‘address’ domains, in particular localisation and secretion signals.

Basic assembly and gene calling

We take as our example the production and analysis of a whole organism gene complement. The production of this dataset requires first assembling a draft genome and identifying the genes, or de novo assembly or mapping of transcriptome sequence data. The MIRA assembler (Chevreux, Wetter & Suhai, 1999) supports multiple sequencing technologies and is capable of assembling viral, bacterial and smaller eukaryotic genomes. We wrapped MIRA for use within Galaxy as a useful tool for pathogen sequence assembly. However, MIRA’s high memory requirements on larger genomes may require cluster configuration to adjust job scheduling. As is the case for many assemblers, the primary output file of interest is a FASTA format file of assembled sequence contigs.

A basic workflow for MIRA transcriptome assembly in Galaxy is: • Upload or import sequencing read data in FASTQ format (Cock et al., 2010).

• Run “Assemble with MIRA v3.4” in transcriptome mode to produce assembled mRNA sequences in FASTA format.

• Run “Get open reading frames (ORFs) or coding sequences (CDSs)” (or the EMBOSS tool getorf (Rice, Longden & Bleasby, 2000) which is also available in Galaxy) to obtain a set of putative protein sequences.

This simple protocol does not consider complications such as frame-shifts due to sequencing or assembly errors, but represents a simple initial framework to begin such an analysis.

The eukaryotic gene caller Augustus (Keller et al., 2011), and a prokaryotic gene caller, Glimmer3 (Delcher et al., 2007) were both wrapped in Galaxy to provide tools for extracting candidate gene sequences from genomic assemblies. Both tools accept FASTA format sequence contigs as input, such as those produced by MIRA. The output of both tools can be obtained either in gene transfer format (GTF) or general feature format (GFF). These are standard tabular tab-separated plan text formats, suitable for input and visualisation in common genomic viewers, including Galaxy’s own visualisation tool, Trackster (Goecks et al., 2012). Moreover, the wrappers extract the predicted gene and protein sequences and provide them in FASTA format, for convenience of downstream analysis.

Assembled genomes typically contain large stretches of repetitive and low complexity sequence. Also, many organisms share very similar repetitive sequences, despite being phylogenetically distant. If these sequences are not masked prior to performing sequence similarity database searches or creating models based on sequence, the results may not accurately reflect the true evolutionary relationships between sequences. The RepeatMasker tool searches for interspersed repeats and low-complexity sequence and produces an output sequence where these regions are replaced with ‘N’ ambiguity symbols (Smit, Hubley & Green, 1996–2010). This is accomplished by sequence comparison, searching against a database of canonical repeats. The Galaxy wrapper for RepeatMasker accepts FASTA format nucleotide sequences, and returns a FASTA file of masked sequences. This tool can also provide an HTML (webpage) summary and GFF annotation file of all identified repeats.

Together, these tools can be used in a basic workflow that takes raw sequencing data as input, yielding a whole organism gene set that can be further analysed: • Upload or import sequencing read data in FASTQ format.

• Run “Assemble with MIRA v3.4” in genome mode to produce assembled contig sequences in FASTA format.

• Run “RepeatMasker” on the assembled FASTA file to generate a masked FASTA file.

• Run “Augustus gene prediction for eukaryotic genomes”, or “Glimmer3” for prokaryotes, on the masked FASTA file.

Glimmer may be trained on an existing set of genes from related organisms, to improve the accuracy of de-novo gene prediction for a given organism. Figure 3 shows a more complex workflow which takes as input a set of known gene structures for training. The tool “Glimmer ICM builder” takes this set of structures as input, and creates an interpolated context model (ICM) that is used by the “Glimmer3” tool to predict novel genes on the second input sequence. This example workflow is available on the Galaxy Tool Shed (Table 1).

Figure 3 Screenshot from Galaxy workflow editor illustrating the Glimmer3 gene finding example discussed in “Basic assembly and gene calling”.

Basic annotation using BLAST

A simple justification often used for transfer of functional annotation from a protein of known function to one of unknown function is on the basis of overall sequence similarity. Consequently, BLAST is often one of the first tools used for initial functional annotation of de novo assembled transcripts, or predicted gene calls from a draft genome assembly. Inspection of a table giving descriptions of the top three BLAST matches in the NCBI non-redundant (nr) database to a query protein of unknown function can give rudimentary annotation information, with limited representation of the variation in annotated function, for sequence matches. These basic annotations can be used to explore differential expression related to protein function, as in Palomares-Rius et al. (2012) where preliminary functional annotation was performed as a simple two step job in Galaxy: • Upload or import FASTA file of protein (or transcript) sequences.

• Run “NCBI BLAST+ blastp” (or “blastx”) against the NCBI nr database, requesting XML output and – optionally – only the top three hits.

• Run “BLAST top hit descriptions” requesting the top three descriptions.

The resulting table can be saved locally for import of annotations into another analysis package if desired, such as R/BioConductor (Gentleman et al., 2004) or GeneSpring (Agilent Inc.) for incorporation into expression analysis. Sequence similarity as indicated by a one-way BLAST search alone should not be taken as strong evidence for orthology or common function (Punta & Ofran, 2008). So, while this example does not represent annotation best practise, it is useful as a simple and straightforward first step to explore a new assembly.

Another popular tool for initial functional annotation of novel gene sequences is Blast2GO (Conesa et al., 2005; Götz et al., 2008) which, in addition to its Graphical User Interface (GUI), offers a command line variant b2g4pipe (Blast2GO for pipelines) which we have wrapped for use in Galaxy. We implement whole organism scale Blast2GO in Galaxy as a simple two step pipeline, as used in Palomares-Rius et al. (2012), taking advantage of a local computing cluster: • Upload or import FASTA file of protein (or transcript) sequences.

• Run “NCBI BLAST+ blastp” (or “blastx”) against NR requesting XML output.

• Run “Blast2GO” on this XML file (ideally using a local Blast2GO database for speed).

The resulting tabular annotation file can also be downloaded for import into the Blast2GO GUI tool if desired.

Basic annotation using InterProScan

InterPro is a protein functional annotation resource that combines information from multiple third-party databases to classify proteins into families by predicted domains and active sites (Hunter et al., 2012b). The InterPro consortium incorporates several member databases including: PRINTS, PROSITE, HAMAP, PFAM, TIGRFAM, CATH-Gene3D, ProDom, Superfamily, SMART, Panther and PIRSF. These member databases take complementary approaches to the problem of assigning function to protein sequences. For example, the PRINTS database uses conserved motifs to characterise a protein family or domain, which enables a large number of proteins to be quickly identified with a relatively low computational overhead, whereas in contrast, the PFAM database uses multiple sequence alignments and profile hidden Markov models to define protein domain families. By combining information from each of the member databases, InterPro is able to provide an integrated functional classification.

The InterPro consortium provides a central portal that enables small numbers of protein sequences to be classified. For larger jobs, such as whole organism gene complement functional annotation, the standalone InterProScan tool (Zdobnov & Apweiler, 2001; Quevillon et al., 2005) can be installed locally. We have created an InterProScan wrapper which enables this application to be run as part of the Galaxy environment. The wrapper accepts FASTA format protein sequences as input, and returns the results as tabular data. Given a transcriptome or EST assembly, a basic workflow may consist of the two step pipeline: • Upload or import FASTA file of nucleotide sequences.

• Run “Get open reading frames (ORFs) or coding sequences (CDSs)” (or the EMBOSS tool getorf (Rice, Longden & Bleasby, 2000) which is also available in Galaxy) to produce protein sequences.

• Run “InterProScan” on these protein sequences.

Prediction of secreted proteins

The nematode effector finding protocol in Jones et al. (2009) was one of the first workflows we wished to reconstruct in Galaxy, for local use by several research groups. This protocol identifies proteins that possess a signal peptide, but no transmembrane domains, using SignalP 3.0 (Bendtsen et al., 2004) and TMHMM 2.0 (Krogh et al., 2001). The authors of these tools offer webservices for each, but these are not readily integrated together, or into an automated analysis such as a Galaxy workflow. They are also limited in both the number and length of sequences they will accept as input. Both SignalP and TMHMM were wrapped for Galaxy, and the sequence filtering tool described above written to enable the implementation of a simplified version of the method in our local Galaxy environment (Cock & Pritchard, in press). This workflow is available to download and install from the Galaxy Tool Shed (Table 1), and has been applied to a number of plant pathogens, including other nematodes (Kikuchi et al., 2011; Kikuchi et al., in press), bacteria, aphids, oomycetes and fungi.

Simple comparison of RXLR predictions

Many oomycete plant pathogens produce effector proteins that share a common RXLR translocation motif, and several methods have been proposed to predict the presence of this motif (Bhattacharjee et al., 2006; Win et al., 2007; Whisson et al., 2007). We implemented three different published RXLR prediction methods as a new tool for Galaxy (see Methods), and wished to compare directly the agreement between these approaches. Figure 4 shows a simple workflow which takes as input a multiple protein sequence FASTA file, carries out each of the three RXLR prediction methods, filters their output for positive matches only, and displays the results as a Venn Diagram. An extended version of this workflow is available to download and install from the Galaxy Tool Shed (Table 1), which also generates a FASTA file of those proteins with a positive prediction from all three methods (Cock & Pritchard, in press). Production of a PDF Venn Diagram like this is a quick and easy way to obtain a visual summary of the overlap between a small number of classification processes – for example differentially expressed genes from an RNASeq pipeline, or a microarray analysis imported into Galaxy.

Figure 4 Screenshot from Galaxy workflow editor illustrating the simple RXLR and Venn Diagram example discussed in “Simple comparison of RXLR predictions”.

Other classification tools

Many of the existing tools we have wrapped for use in Galaxy are for protein sequence analysis, and have the aim of predicting protein localisation. The PSORT family of tools implements protein localization prediction for a range of organisms. PSORTb is applicable to bacterial and archaeal proteins (Yu et al., 2010), and WoLF PSORT is intended for eukaryotic proteins (Horton et al., 2007). Predictions of mature effector protein host localisation can suggest possible modes of action, and may be used to prioritise candidates for further analysis by methods such as in situ confocal microscopy. Many pathogen effector proteins target the host nucleus, presumably for the purpose of modifying or disrupting DNA regulatory processes. We were therefore interested in plant nuclear (NLS) and nucleolar (NoLS) localization sequence prediction. Most published tools in this area were specialised to known signals from human sequences. Nevertheless, we implemented wrappers for the NLS prediction tools NLStradamus (Nguyen Ba et al., 2009) and PredictNLS (Cokol, Nair & Rost, 2000), and the NoLS detector NoD (Scott et al., 2010; Scott, Troshin & Barton, 2011).

In order to predict the presence of bacterial type III secretion signals in the Galaxy framework, we wrapped the EffectiveT3 tool (Arnold et al., 2009; Jehl, Arnold & Rattei, 2011). This package aims to identify a specific secretion signal in the N-terminal region of bacterial proteins. An alternative approach to identification of type III effectors, taking into account genomic context, would be to identify the distinctive HrpL alternative sigma factor binding site often found upstream of type III effector genes (Yang et al., 2010; Baltrus et al., 2011; McNally et al., 2011). In general, expression in parasitic life stages is a useful criterion for identifying candidate effector genes, and so tools for the prediction or identification of promotor binding sites are an area we hope to incorporate into our Galaxy setup. Currently, we have only wrapped Promoter 2.0 (Knudsen, 1999), which is specific to identification of eukaryotic PolII promoters.

Conclusions

In discussions to plan provision of local access to bioinformatic tools, databases, and analysis scripts at the authors’ host institutes, a key requirement was that any solution should be extendable, with the ability to add new tools. Web-based solutions were particularly appealing as these involve no end user software installation, other than a recent standards-compliant web-browser, which is present by default on any recent operating system. It was recognised that increasing computational demand for processing sequence data was already limiting the scope for effective computation on many users’ own desktop machines. Hence, a client–server architecture able to take advantage of the existing computing cluster was preferred. This has had the additional benefit of increasing use of the institute’s cluster and associated storage, helping justify upgrades to these resource as being of benefit beyond only computational biologists.

In selecting Galaxy, several alternative frameworks were considered and rejected, including both commercial offerings with concerns about vendor lock-in and price, and other open source projects taking a different approach. For example, Taverna required the user to install Java client software, and is heavily orientated towards web-services (Wolstencroft et al., 2013). A possibility we overlooked at the time is Yabi (Hunter et al., 2012a).

As discussed, Galaxy fulfills our core requirement of being extensible to add new tools. Most of the examples discussed herein wrap existing third party software, but some are based around locally-developed scripts such as the RXLR motif finder. A local Galaxy server provides a relatively easy way for a core bioinformatics facility or an embedded bioinformatician to deliver convenient access to locally-produced bespoke analysis scripts or tools, avoiding the substantial effort needed to provide a GUI or separate web-interface, or to install or maintain the tool for each individual user. If the tool author wishes to share their work via the Galaxy Tool Shed, it is now possible to set up automated installation of dependencies, subject to licensing constraints. In an ideal case, such as for the NCBI BLAST+ wrappers, the Galaxy administrator can install a new tool and its dependencies from the Galaxy Tool Shed, and run the unit tests, via a simple point-and-click process.

We have found Galaxy very useful to make tools and associated data available within our institute, echoing experiences described at other sites (Maclean & Kamoun, 2012). In particular, we have found that this has enabled a productive mode of working where a bioinformatician can develop a workflow (or new tool) in discussion with wet-lab biologists, and then make this workflow available to them, and the wider user base, to use on demand. Constructing common workflows in this way ensures that methodology is standardised and documented, and an automated workflow additionally reduces the scope for human error. The tools or workflows described have already been used in a number of plant pathology publications, as noted above (Palomares-Rius et al., 2012; Kikuchi et al., 2011; Kikuchi et al., in press), and in other recently submitted work. Galaxy workflows can be included with publications, either hosted on the Galaxy Tool shed or included as a Supplemental Information. This allows protocols to be run on other Galaxy servers (provided that the underlying tools have also been installed) and encourages reproducible science.

We thank the Galaxy Team for their mailing list support, and in particular Dannon Baker for migration of the BLAST+ tools to the Galaxy Tool Shed. Edward Kirton is thanked for his input in handling BLAST databases, and the following for assistance in wrapping their tools in Galaxy: Alex Nguyen Ba (NLStradumus author), Peter Troshin (NoD co-author), and Laszlo Kajan (PredictNLS maintainer). The Academic Editor for PeerJ, Gerard Lazo, and the two reviewers, Mick Watson and Mikel Egaña Aranguren, are thanked for their constructive feedback. We also thank the users of our local Galaxy installation at the James Hutton Institute and Iain Milne, as this work would not have happened without their enthusiasm and support.

Additional Information and Declarations

Competing Interests

Author Contributions

Data Deposition

The authors declare that they have no competing interests.

Peter J.A. Cock, Björn A. Grüning and Konrad Paszkiewicz conceived and designed the experiments, performed the experiments, analyzed the data, contributed reagents/materials/analysis tools, wrote the paper.

Leighton Pritchard conceived and designed the experiments, contributed reagents/materials/analysis tools, wrote the paper.

The following information was supplied regarding the deposition of related data:

The tools and workflows described have been deposited in the Galaxy Tool Shed, links given in Table 1.

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
