# Peer review of "Galaxy tools and workflows for sequence analysis with applications in molecular plant pathology"

_PeerJ, doi:10.7717/peerj.167_

## Round 0.1 · original submission · Minor Revisions

The manuscript appears well written and is poised to provide Galaxy work-flows for bench scientists wishing to conduct transcript and peptide analyses for plant pathology related studies. Both reviewers felt the manuscript was appropriate for publication with what I consider to be minor modifications. The Galaxy environment has gained wide acceptance and I feel your application topic may help aid analyses in other plant pathology systems. This would potentially lead toward building common data connections between diverse host-pathogen interactions, and may even extend beyond the limited area of focus. Your presentation is centered around plant pathology based studies; however, it mainly focuses on the tools and not the research findings. Given that the introduction went well into describing the importance of plant pathology studies, perhaps a mention of some plant pathology revelations uncovered from your work may strengthen the impact of this effort. I will forward this to you with a suggestion of minor modifications. I would like you to try to address the points suggested in the reviews; it does not seem to be a major hurdle to accomplish this in a short period of time. Thank you for submitting this manuscript and I expect it to be well received. Congratulations on your efforts.

Other comments which may be useful are to mention software alternatives to Galaxy and to mention to what extent the tools contained within the work-flows can also be used via the command-line. When mentioning third-party software it is best to note their availability; whether a license is required or not. Since the target audience appears the general bench scientist a description of the system requirements in terms of memory and processor requirements would be helpful. A sample data-set may also serve to let the target audience use and test for the expected outcomes.

Additional edits suggested :
Example of annotation:
LINE NO.: / PREVIOUS FORM / SUGGESTED FORM / [ADDITIONAL NOTES]

68: / Apple’s Mac OS X / the Apple OS X / [Mac is semi-redundant; see wikipedia]
78: / to support / for support / []
84: / offers is to offer / offers is / []
114: / server can made / server can be made / []

·

Basic reporting

The paper meets the requirements

Experimental design

The paper meets the requirements

Validity of the findings

The paper meets the requirements

Additional comments

The authors describe a number of tools and tool wrappers that have been integrated into Galaxy, and provide a use-case in molecular plant pathology

There could be more mention of alternatives to Galaxy, e.g. Taverna and Anvaya

Whilst MIRA has been integrated, no mention is used of the memory requirements - many are reluctant to integrate assemblers into their Galaxy instances for fear that several large memory jobs are launched by users

On page 5, two workflows are mentioned that are essentially identical, except one uses GetOrfs for gene finding and the second uses Augustus and Glimmer3. Doesn't the second workflow make the first redundant? Why include the first?

On page 6, technically I feel orthology should be the basis for transferring functional information, not sequence similarity. Similarly on page 7, isn't it more standard to use reciprocal best hit to define orthologues before transferring annotation?

Bottom of page 7, GetOrfs is used again - why not use the aforementioned gene predictors?

Was any attempt made to wrap the InterProScan web-service (rather than standalone)?

Top of page 8, I am curious whether the SignalP licence allows for it to be integrated into a public Galaxy?

The RXLR prediction tools: as I understand it, the authors have implemented several published methods for RXLR motif prediction, and released these into the Galaxy tool shed. Does this paper serve as notice of their publication? Has any testing been done on these implementations to demonstrate their accuracy and efficacy?

Overall the paper is well written and should be published. The above suggestions can be dealt with by adding text to various parts of the manuscript and do not represent a large body of work, therefore I recommend minor revisions

Mick Watson

·

Basic reporting

The paper is very well written and presents the ideas clearly.

Some minor (Discretionary) comments regarding the style:

* The title is too long, how about "A Galaxy framework for sequence analysis with applications in molecular plant pathology"?.
* In the abstract, NCBI BLAST+ is mentioned and then BLAST is mentioned again, but as an example. It id confusing.
* In the abstract, in the sentence "The motivating research theme ... " it is not clear whether the research theme mentioned refers to Galaxy as a whole or the content of this paper. Also, the abstract reads like a presentation of Galaxy, rather than presenting the authors' work (Specific Galaxy tools).
* The sentence in lines 148-151 is very difficult to understand.
* The last part of the sentence in lines 244-245 may be clearer written as follows: "despite being phylogenetically distant"

Possible mistakes:

Line 112: computING cluster?
Line 114: can BE made
Line 115: extra space after "e.g."? Perhaps the authors can use the LaTex command \newcommand{\eg}{\emph{e.g.}\xspace} (and the xspace package)
Line 244: sequenceS

Experimental design

The main objection is that the work presented in this paper is not completely reproducible.

The authors present a set of Galaxy tools and workflows that exploit such tools. However, only the "backbones" of the workflows are stored in the Galaxy tool shed. Therefore, if a user wants to reproduce the workflow, she needs to import it into a Galaxy server and run the workflow with datasets of her choice: since the datasets will be different, the workflows are not completely reproducible.

The authors should publish the workflows with the datasets they used to test them. Since the authors mention in the acknowledgements that they maintain an in-house Galaxy server, they can easily make the workflows mentioned in the paper public, and also publish a history with the datasets used, with clear instructions mapping the datasets to the corresponding workflow steps. This way any reader can run precisely the workflows presented in the paper, with the actual datasets, and judge the results. If the authors are worried about the computational burden for their server, they can set up accounts for the reviewers only, without making their Galaxy server public.

Validity of the findings

As already mentioned, the datasets used to test the workflows have not been made available.

---

## Round 0.2 · accepted · Accept

Thank you for your feedback on the suggested revisions. My read-through went very smoothly and I feel you have addressed all concerns addressed in the initial review. The manuscript was in good shape in the first iteration of review, and it is now even better. Having worked in the area of plant pathology I feel this work can have impact in serving the newest concerns of the science field, and especially to serve well with the latest advances in technology. I applaud your efforts and I expect the feedback to match accordingly when published. Congratulations.